# Comparative Study of DNA Extraction Methods for the PCR Detection of Intestinal Parasites in Human Stool Samples

**DOI:** 10.3390/diagnostics12112588

**Published:** 2022-10-25

**Authors:** Siriporn Srirungruang, Buraya Mahajindawong, Panachai Nimitpanya, Uthaitip Bunkasem, Pattama Ayuyoe, Surang Nuchprayoon, Vivornpun Sanprasert

**Affiliations:** 1Lymphatic Filariasis and Tropical Medicine Research Unit, Chulalongkorn Medical Research Center (Chula-MRC), Faculty of Medicine, Chulalongkorn University, Bangkok 10330, Thailand; 2Department of Parasitology, King Chulalongkorn Memorial Hospital, The Thai Red Cross Society, Bangkok 10330, Thailand; 3Division of Dermatology, Department of Internal Medicine, Faculty of Medicine, Chulalongkorn University and King Chulalongkorn Memorial Hospital, The Thai Red Cross Society, Bangkok 10330, Thailand; 4Department of Obstetrics and Gynecology, Faculty of Medicine, Chulalongkorn University and King Chulalongkorn Memorial Hospital, The Thai Red Cross Society, Bangkok 10330, Thailand; 5Department of Parasitology, Faculty of Medicine, Chulalongkorn University, Bangkok 10330, Thailand

**Keywords:** DNA extraction, bead-beating procedure, stool samples, intestinal parasites, polymerase chain reaction, PCR inhibitor

## Abstract

Stool samples typically contain PCR inhibitors; however, helminths are difficult to lyse and can cause false-negative PCR results. We assessed the effective methods for extracting DNA from different kinds of intestinal parasites. We compared the most common DNA extraction methods from stool samples, including the phenol-chloroform technique with or without a bead-beating step (P and PB), a QIAamp Fast DNA Stool Mini Kit (Q), and a QIAamp PowerFecal Pro DNA Kit (QB). Genomic DNA was extracted from 85 stool samples collected from patients infected with *Blastocystis* sp., *Ascaris lumbricoides, Trichuris trichiura,* hookworm, and *Strongyloides stercoralis*. DNA quantity and DNA quality were evaluated via spectrophotometry, and DNA integrity was assessed by PCR. We found that P and PB provided higher DNA yields (~4 times) than when using Q and QB. However, P showed the lowest detection rate of PCR (8.2%), wherein only *S. stercoralis* (7 out of 20 samples) was detected. QB showed the highest detection rate of PCR (61.2%). After plasmid spikes, only 5 samples by QB were negative while 60 samples by P were still negative. Remarkably, QB could extract DNA from all the groups of parasites that we tested. These results indicate that QB is the most effective DNA extraction method for the diagnosis and monitoring of intestinal parasites via PCR.

## 1. Introduction

Intestinal parasitic detection methods in routine practice are based on conventional methods, which comprise fecal simple smear microscopy, the formol-ethyl acetate concentration technique (FECT), fecal flotation, a Kato–Katz thick smear, and fecal cultivation. Fecal simple smear microscopy is the most convenient method and consumes the least time when compared to other methods. FECT helps to improve sensitivity for microscopic examination [1]. The Kato–Katz thick fecal smear is the most common method recommended by the World Health Organization (WHO) [2]. However, this method has its limitations, especially its low sensitivity to mild infections in individuals and low sensitivity when using a single smear; it also has poor sensitivity for *S. stercoralis* infections. Moreover, these methods still have much lower sensitivity than cultivation techniques [1,2,3]. Despite being considered the gold standard for detection, cultivation requires a fresh stool with viable parasites and needs more time for analysis than other detection methods. Additionally, the intermittent excretion of eggs/larvae, as well as the numbers of excreted eggs/larvae have been shown to vary, depending on the species of the parasites. These issues result in low sensitivity of detection and might cause false negative results from microscopic examination [2], leading to the underestimation of the disease burden.

Immunological techniques for antigen or antibody detection have been reported to be useful for the diagnosis of parasitic infection. These techniques have a higher sensitivity compared to other conventional parasitological tests. However, they do lack specificity. Cross-reaction with other parasites has frequently been reported. Moreover, antibodies tend to persist for a long time after treatment and cannot be used to indicate active infections [2].

In recent decades, molecular diagnostic methods have been prioritized. PCR-based assays also allow downstream applications, such as species and strain differentiation, epidemiological studies, DNA cloning, and protein expression. However, there are several reports that show false negative results from PCR, even when using DNA extracted from positive samples via microscopy [4,5,6]. These results might be caused by the strong eggshells and the hard and sticky cuticles of the helminths, debris, and fibers in the stool samples. Not only DNA inhibitors but also various PCR inhibitors have been found in stool samples [7]. The quality and quantity of PCR inhibitors vary between samples [7], depending on clinical, dietary, gut microbiota, or other factors in terms of environment and lifestyle. Therefore, DNA extraction methods are crucial to extract a sufficient quantity and quality of genetic materials from stool samples, especially when various types of organism species need to be detected.

DNA extraction methods have been developed to improve the efficiency of separating parasites from stool debris, to break the strong eggshell or cuticle of parasites, and to reduce the components of PCR inhibitors in stool samples. Several pretreatment procedures have been applied, based on chemical [8], thermal [9], and mechanical procedures [7]. The bead-beating method is the most common mechanical procedure for improving DNA extraction efficiency and increasing DNA recovery, both in specific organism studies [7,10] and in microbiome studies [11,12]. Many DNA extraction kits from stool samples are commercially available, using different lysis methods. The use of different DNA extraction kits can deliver a different DNA yield, PCR results, and organism DNA composition for the same stool sample [13,14]. Most comparative studies that assess the DNA extraction method focus on only the specific organism [4,5,6,8,9,10]. However, for disease diagnosis, the selection of a suitable DNA extraction method is crucial to extract the wide range of unknown organisms in the clinical samples. 

Therefore, in this study, we assessed the most effective DNA extraction method for extracting DNA from the various types of common intestinal parasites in stool samples. The *Blastocystis* sp. was selected as the most fragile protozoa, while *Ascaris lumbricoides* has the strongest eggs, and *S. stercoralis* presents the larval stage with the strongest cuticle. Moreover, hookworm and *T. trichiura* were also included as the most common intestinal parasites found in Thailand. We compared 4 methods of genomic DNA extraction from stool samples for the detection of various intestinal parasites using PCR, including the conventional phenol-chloroform technique (P), the modified phenol-chloroform technique with glass beads (PB), the QIAamp Fast DNA Stool Mini Kit (Q), and the QIAamp Power Fecal Pro DNA Kit (QB). Spectrophotometry was performed to evaluate DNA quality and DNA quantity, while DNA integrity was assessed via PCR. Furthermore, the samples that were negative in the PCR were further assessed by spike tests—adding a known amount of plasmid DNA harboring a specific target gene into the extracted DNA samples—to evaluate the presence of PCR inhibitors after the process of DNA extraction.

## 2. Materials and Methods

### 2.1. Stool Sample Collection

A total of 85 stool samples were collected from patients infected with intestinal parasites, including *Ascaris lumbricoides* (*n* = 9), *Trichuris trichiura* (*n* = 5), *Strongyloides stercoralis* (*n* = 20), hookworm (*n* = 14), and *Blastocystis* sp. (*n* = 37), in Chiang Mai Province and Lampang Province in Thailand. The presence of the eggs, larvae, or cysts of parasites was confirmed via simple smears and the formalin ethyl acetate concentration technique (FECT), as described previously [1,15,16]. After performing microscopic examinations, approximately 2 g from each stool sample was preserved in 5 mL of 70% ethanol for DNA extraction. The remaining stool samples were kept at room temperature during transportation to the Department of Parasitology, Faculty of Medicine, Chulalongkorn University, to undergo agar plate culture and LES culture. The sample was considered positive if either the microscopy or culture was positive.

All stool samples in 70% ethanol were kept at −20 °C and were washed three times with sterile distilled water, before undergoing DNA extraction. Four aliquots of 200 mg (or 0.2 mL) of each stool sample in 2 mL sterile microcentrifuge tubes were prepared for performing 4 different DNA extraction methods, including the phenol-chloroform technique (P), the modified phenol-chloroform technique with glass beads (PB), the QIAamp Fast DNA Stool Mini Kit (Q) (QIAGEN, Hilden, Germany), and the QIAamp Power Fecal Pro DNA Kit (QB) (QIAGEN, Hilden, Germany).

### 2.2. DNA Extraction by the Phenol-Chloroform Extraction Method

DNA extraction by the conventional phenol-chloroform extraction method (P method) was performed as described previously [16]. Briefly, lysis solution (20 mM Tris-HCl pH 7.6, 2.5 mM MgCl_2_, 50 mM KCl, 150 µg/mL proteinase K, 0.5% Tween-20) was added to a microcentrifuge tube containing 200 mg (or 0.2 mL) of stool sample, mixed by vortexing until the stool became homogeneous. The sample tubes were then incubated at 65 °C for 3 h and continued at 90 °C for 10 min to inactivate the proteinase K. After that, 200 μL of phenol:chloroform:IAA (25:24:1) was added and mixed, followed by centrifugation at 13,000 rpm at 4 °C for 10 min. The upper aqueous part was then transferred to a new microcentrifuge tube. Two volumes of chloroform were added, mixed thoroughly by inverting, and centrifuged at 13,000 rpm at 4 °C for 10 min. Again, the upper part of the supernatant was transferred into a new microcentrifuge tube; we then added 2.5 volumes of ice-cold absolute ethanol and 0.1 volume of 3M sodium acetate (pH 5.2). All samples were kept at −20 °C overnight to precipitate DNA. The precipitated DNA was collected by centrifugation at 13,000 rpm at 4 °C for 10 min. The DNA pellet was washed with 1000 μL of 70% ethanol and air-dried at room temperature. Finally, 100 μL of TE buffer was used to re-suspend the DNA pellet.

For the modified phenol-chloroform extraction method with glass beads (PB method), we added the bead-beating pretreatment step by mixing 200 mg of stool samples with 250 mg of sterile 0.5 mm glass beads (Omni International, Kennesaw, GA, USA) in 400 μL of lysis solution. The mixture was horizontally vortexed at maximum speed for 10 min until the stool became homogeneous. The following steps were performed as described above.

### 2.3. DNA Extraction by the QIAamp Fast DNA Stool Mini Kit

DNA extraction was performed as recommended by the manufacturer (QIAGEN, Hilden, Germany) with few modifications to ensure that we could gain the maximum yield from each stool sample. One milliliter of InhibitEX buffer was added to the tube containing 200 mg (or 0.2 mL) of stool sample. The stool was mixed by a vortex, and we made sure that the stool was completely removed from the sides of the microcentrifuge tube. The lysis temperature was maintained at 95 °C for 5 min to ensure the lysis of parasites in the stool samples, as recommended by the manufacturer. The suspension was centrifuged at full speed for 1 min to pellet the stool particles; the upper supernatant was collected to mix with the proteinase K in a new tube. Then, 200 μL of buffer AL was added and mixed by vortexing for 15 s. The supernatant was incubated at 70 °C for 10 min. Absolute ethanol (200 μL) was then added to the lysate. The lysate was then carefully applied to the QIAamp spin column and centrifuged at full speed for 1 min. The column was washed 2 times with washing buffers (Buffer AW1 and AW2, respectively). Then, a total of 100 μL of elution buffer (Buffer ATE) was added and centrifuged at full speed for 1 min to elute the DNA.

### 2.4. DNA Extraction by the QIAamp Power Fecal Pro DNA Kit

DNA extraction was performed following the manufacturer’s recommendations (QIAGEN, Hilden, Germany) with few modifications to compare the DNA extraction efficiency to the other methods. DNA was extracted from 200 mg (or 0.2 mL) of stool sample. The stool sample and 800 μL of lysis buffer (solution CD1) were added to the PowerBead Pro tube. The suspension was horizontally vortexed at maximum speed for 10 min until the stool became homogeneous. The lysate was then centrifuged at 15,000× *g* for 1 min. The supernatant was collected and mixed with 200 μL of solution CD2, which contained inhibitor removal reagents. After centrifugation, the supernatant was collected and mixed with 600 μL of Solution CD3, which adjusted the DNA solution salt concentration and allowed more specific DNA binding to the column. The suspension was then carefully applied to the QIAamp spin column and centrifuged at full speed for 1 min. The column was washed 2 times with washing buffers (solutions EA and C5, respectively). Then, a total of 100 μL of elution buffer (Solution C6) was added and centrifuged at full speed for 1 min to elute the DNA.

### 2.5. Quantitative and Qualitative Assessment of Extracted DNA

The concentration and purity of the extracted DNA were determined using a Nanodrop 1000 spectrophotometer (Thermo Scientific, Waltham, MA, USA). The measurement was repeated by taking triplicates of each sample. DNA purity was assessed by the OD260/280 absorbance ratios for protein contamination, and by the OD260/230 absorbance ratios for polysaccharides, phenols, chloroform, and other contaminations. The percentage of DNA samples with optimum purity (OD260/280 = 1.80–2.00, and OD260/230 = 2.0–2.2) was determined. To evaluate the integrity and length of purified DNA fragments, 50 ng of each DNA sample was analyzed via electrophoresis on 1% agarose gel. The extracted DNA samples were stored at −20 °C until we performed PCR.

### 2.6. Polymerase Chain Reaction (PCR)

The small subunit ribosomal DNA (SSU rDNA) and ITS1 for *A. lumbricoides*, *N. americanus, S. stercoralis,* and *Blastocystis* sp. were amplified, as described previously [15,16,17]. Specific primers for *T. trichiura* were designed using the Primer 3 program. The primer sequences and product sizes are described in Table 1.

The PCR was conducted in a 25-μL reaction containing 1 × PCR buffer, 3 mM of MgCl_2_, 0.2 mM of each dNTP, 1U of *Taq* DNA polymerase (Fermentas, Waltham, MA, USA), 40 µg/mL of BSA, 0.4 µM of each primer, and 200 ng of the DNA template. The amplification program included the pre-heating denaturation at 95 °C for 4 min, 35 cycles of denaturation at 95 °C for 30 s, annealing at 55.4 °C for 30 s, and extension at 72 °C for 30 s. The final amplification cycle included the addition of a 4-min extension at 72 °C. PCR products were analyzed via 1% agarose gel electrophoresis in 1 × TAE buffer for 30 min. The gel was stained with ethidium bromide to visualize the DNA fragments under a UV transilluminator. The PCR products of each parasite were cloned into pGEM^®^-T Easy Vector (Promega, Madison, WI, USA), as described previously [16]. These plasmids were used as “plasmid controls” in PCR and in the spike tests.

### 2.7. Assessment of PCR Inhibitors in DNA Extracted from Stool Samples

The presence of PCR inhibitors in the extracted DNA was evaluated via plasmid spike tests. All PCR-negative DNA samples were directly spiked with 0.1 ng of pGEM^®^-T Easy Vector, harboring fragments of specific parasites (2.86 × 10^7^ copies of ITS1 of *A. lumbricoides*, 2.65 × 10^7^ copies of 18S rDNA of *N. americanus*, 2.54 × 10^7^ copies of 18S rDNA of *T. trichiura,* 2.97 × 10^7^ copies of 18S rDNA of *S. stercoralis*, or 2.65 × 10^7^ copies of 18S rDNA of *Blastocystis* sp.). Then, PCR amplification of the specific gene was performed, as described above. Positive controls in the inhibition test via spiking were of the same plasmid that we spiked into the DNA samples. If the amplicons of targeted genes were observed, no PCR inhibitors in the DNA samples were suspected. The plasmid controls were used as a positive control. 

### 2.8. Data Analysis

Data were recorded and analyzed using the Microsoft Excel 2010 program and GraphPad Prism software, version 9.4.1 (GraphPad Software Inc., San Diego, CA, USA). Differences between the DNA extraction methods in terms of DNA concentration were determined using the Wilcoxon matched-pair signed rank test. To compare DNA purity between groups, we performed estimation plots and applied a paired *t*-test. The PCR detection rates with the methods were compared using the chi-square test. A *p*-value of < 0.05 was considered to be statistically significant.

## 3. Results

### 3.1. DNA Concentration of the Extracted DNA

The conventional P method provided a significantly higher DNA yield than the Q method (geometric mean = 165.68 and 46.30 ng/μL, respectively) (*p* < 0.0001) (Figure 1A). According to the estimation plot, the mean of the difference between the concentration of DNA extracted by P and Q was 189.4 (ranging from −1082.8 to 89.5 ng/μL) (Figure 1C). 

There were 11 out of 85 samples (12.9%) that showed a lower DNA yield by the P method. All these 11 samples had DNA concentrations extracted by the P test of lower than 100 ng/μL (Figure 1D). The maximum DNA concentration provided by the P method was 1194.4 ng/μL, while the maximum DNA concentration provided by the Q method was only 222.9 ng/μL (Figure 1D). These might be due to the maximum binding capacity of the QIAGEN column.

The addition of the bead-beating pretreatment step in the P method (PB) could significantly increase the DNA yield (geometric mean = 205.54 ng/μL) (*p* < 0.0001). In contrast, the concentrations of DNA extracted by the Q and QB methods were not significantly different (geometric mean = 46.30 and 53.76 ng/μL, respectively) (*p* > 0.05) (Figure 1A). The mean of the difference between P and PB was 45.92 (ranging from −365.6 to 520.2 ng/μL) (Figure 1B), while the mean of the differences between Q and QB was only 9.2 (ranging from −126.6 to 162.4 ng/μL) (Figure 1C). The DNA concentration of the extracted DNA using each method was not correlated with the number of eggs or larvae in stool samples (data not shown). Moreover, we did not identify a correlation between the DNA concentration of extracted DNA by each method.

### 3.2. Quality of Extracted DNA

According to the A260/280 ratio, Q provided the highest ratio of A260/280 (geometric mean = 2.01) (Figure 2A). DNA extracted by the QB method could not increase the A260/280 ratio. The geometric mean of the A260/280 ratio of DNA extracted by the QB method was significantly lower than by the Q method (geometric = 1.89 and 2.01, respectively) (*p* < 0.05) (Figure 2A,C).

Unfortunately, the DNA extracted using the P method was impure. The geometric mean of the A260/280 ratio of DNA extracted by the P method was significantly lower than by the Q method (geometric = 1.66, and 2.01, respectively) (*p* < 0.0001) (Figure 2A,D). Although we added the bead-beating step, the A260/280 ratio of DNA extracted with the PB method was not significantly different from extraction with the P method (geometric = 1.68 and 1.66, respectively) (*p* > 0.05) (Figure 2A,B).

For the secondary measurement of DNA purity, QB provided the highest ratio of A260/230 (geometric mean = 1.62) (Figure 3A). DNA extracted via the QB method showed a significantly higher A260/230 ratio than by the Q method (geometric mean = 1.62 and 1.10, respectively) (*p* < 0.01) (Figure 3A,E). Unfortunately, DNA extracted using the P method was impure. The geometric mean of the A260/230 ratio of DNA extracted by the P method was significantly lower than by the Q method (geometric = 0.58, and 1.10; respectively) (*p* < 0.0001) (Figure 3A,D). Although we added the bead-beating step, the A260/230 ratio of DNA extracted by the PB method was still low (geometric mean = 0.63) (Figure 3A) and was not significantly different from extraction using the P method (*p* > 0.05) (Figure 3B).

Moreover, the percentage of DNA samples with an optimum purity was also determined. While the A260/280 ratio of 1.8–2.0 is generally accepted as the optimum purity of DNA, the expected A260/230 ratio for optimally pure DNA is commonly within the range of 2.0 and 2.2. The P and Q methods provided a comparable number of optimally pure DNA (23 and 25 samples, respectively. The QB method offered the highest number of DNA samples of optimum purity (41 of 85, 48.2%)) (Figure 4). For the A260/230 ratio, QB also gave the highest number of DNA samples of optimum purity (15 of 85, 17.6%), while the P method provided only 5 DNA samples of optimum purity (Figure 4). 

To evaluate the integrity and length of the purified DNA fragments, gel-electrophoresis of the DNA samples was performed (Figure 5). The high molecular weight bands were observed in all methods. Using the P method, more smeared DNA bands were observed, compared to the Q method. The bead-beating pretreatment resulted in increased smeared bands in both the PB and QB methods, compared to the P and Q methods, respectively. 

### 3.3. Detection of Intestinal Parasites by PCR

The efficiency of the various DNA extraction methods was evaluated by comparing the rates of positive PCR values for the intestinal parasites. The different DNA extraction methods showed differences in efficiency regarding the detection of the different parasites (Figure 6). Almost all DNA samples extracted using the conventional P methods were negative for PCR. Only 7 out of 20 (35%) samples from patients infected with *S. stercoralis* were positive, while all samples from patients infected with *Blastocystis* sp., *A. lumbricoides*, *T. trichiura*, and hookworm were negative (Table 2). Adding glass beads in the P method (PB) could detect more than 2 samples with *S. stercoralis* (9 of 20; 45%). Moreover, 5 samples with *Blastocystis* sp. (13.5%) and 1 sample with *A. lumbricoides* (11.1%) were also positive for PCR. However, 70 samples were still negative. The PCR detection rate by the PB method (17.6%) was not significantly different from that of the P method (8.2%) (*p* > 0.05) (Figure 7).

Using the Q method, DNA samples extracted from patients infected with *Blastocystis* sp. (40.5%), *T. trichiura* (20%), hookworm (28.6%), and *S. stercoralis* (75%) were positive (Table 2). The PCR detection rate using the Q method was significantly higher than when using the P method (41.2% and 8.2%; respectively) (*p* < 0.01) (Figure 7). However, none of the DNA samples from patients infected with *A. lumbricoides* could be extracted via the Q method (Table 2). The QB method showed the highest efficiency regarding extracted DNA from all the species of parasites that we examined (Table 2). DNA extracted by QB from patients infected with *Blastocystis* sp. (59.5%), *A. lumbricoides* (77.8%), *T. trichiura* (60%), hookworm (35.7%), and *S. stercoralis* (75%) were positive (Table 2). The PCR detection rate by the QB method (61.2%) was significantly higher than by the Q method (41.2%) (*p* < 0.01) (Figure 6). Unfortunately, 33 out of 85 samples (38.8%) were still negative for PCR using DNA extracted by the QB method (Table 2).

### 3.4. Assessment of PCR Inhibitors in the Extracted DNA Samples

PCR-negative samples were spiked with the plasmid DNA harboring the targeted gene of each parasite, to investigate the PCR inhibitors that contaminated the samples. Among the DNA extracted using the P method, only 18 out of 78 PCR-negative samples (23%) were positive after spiking the plasmid. Sixty DNA samples (70.6%) were still negative, indicating the contamination of PCR inhibitors in the samples (Table 3). Adding glass beads in the P method (PB) could not significantly increase the positive rates by PCR. From 70 PCR-negative samples (*p* > 0.05), only 19 samples (27.1%) extracted by the PB method turned out to be positive by the PCR test, but 51 samples (60%) were still negative. The inhibition rates and the rates of negative PCR results using DNA extracted by the P and PB methods were significantly correlated (*p* < 0.0001).

Using the Q method, the positive rates by PCR were increased compared to when using the P method (*p* < 0.01). From 50 PCR-negative samples, 29 samples (58%), extracted by the Q method, were considered positive but 21 samples (24.7%) were still negative. Remarkably, after the plasmid spikes, 28 of 33 PCR-negative samples (84.8%) that were extracted by the QB method turned out to be positive (Table 3). Only 5 DNA samples (5.9%) were still negative according to the PCR test. These results indicated that the DNA extracted with the QB method showed the fewest PCR inhibitors. Therefore, QB was the most effective method to remove PCR inhibitors contaminated in the stool samples. 

## 4. Discussion

PCR-based techniques offer sensitive, fast, and accurate methods to diagnose parasitic infections. However, PCR results might be affected by several factors, such as the small number of parasites, the intermitted shedding of parasites, and the non-homogeneous nature of the parasites in clinical samples. This results in false-negative PCR results even when the microscopic and/or culture methods are positive. More sensitive molecular methods, such as a nested PCR or real-time PCR, may increase the sensitivity of detection. The targeted genes, the length of amplicons, and PCR conditions, as well as clinical sample storage and preservation, can be crucial for reliable PCR results. However, false-negative PCR results were affected by the presence of numerous PCR inhibitors; DNA degradation during DNA extraction and the poor quality of extracted DNA cannot be solved with the more sensitive techniques. 

Stool samples are commonly contaminated with various kinds of PCR inhibitors that affect the sensitivity of nucleic acid detection [7]. Several substance classes of PCR inhibitors have been widely described, including polysaccharides, fats, glycogen, heme compounds, bile salts metabolic products, phenolic compounds, cellulose, heavy metals, components of bacterial cells, and a large number of non-target nucleic acids [18]. These compounds are associated with differences in clinical, dietary, gut microbiota, or other factors in the environment and lifestyles [18]. The presence of PCR inhibitors in stool samples is also associated with age [19]. PCR inhibition has been reported in none of the stool samples from infants younger than 6 months and is only in 17% of samples from 6- to 24-month-old infants [19]. The occurrence of PCR inhibitors is higher in the adult population compared to infants. The quality and quantity of PCR inhibitors have been reported to be varied between stool samples. Moreover, the variation in the composition and consistency of stool samples from different animal species may affect the quality of the extracted DNA. 

Several approaches have been reported to overcome both DNA inhibition and PCR inhibition, and to facilitate PCR amplification, such as DNA dilution to dilute the inhibitors, chemical additions in reaction conditions, and pretreatment steps before DNA extraction with the same and with other methods [7]. However, the appropriate approaches are varied and are non-specific to the clinical specimens and the target organisms [7]. Previous studies have shown that PCR inhibitors can be bound or partially inactivated by several compounds, such as BSA, formamide, glycerol, and NP-40 [20]. Heating, chromatography, and immunocapture are also effective methods by which to inactivate the PCR inhibitors. However, these methods are time-consuming and might reduce the yield of DNA [18,21]. In this study, we also studied the efficiency of BSA in the inactivation of PCR inhibitors. Some negative PCR samples were tested by adding BSA to the PCR reaction. However, we did not find a positive band after the addition of BSA (data not shown). 

Effective DNA extraction methods are crucial to extracting a sufficient quantity and quality of DNA from stool debris and reducing the various components of the PCR inhibitors. Several DNA extraction methods have been developed to remove or inactivate these PCR inhibitors in the stool samples. However, their efficiency is varied and shows far from complete removal. The phenol-chloroform technique is a conventional DNA extraction method. The lysis buffer used in this method usually contains a detergent for breaking down the cellular membranes. Protease is also added into the lysis buffer for the digestion of protein cellular components. Proteins, carbohydrates, lipids, and cell debris are separated from the extracted DNA by the organic solvent in the process of extraction [18,22]. However, their efficiency varies and shows far from complete removal of the inhibitors. Additionally, the residual amounts of phenol in the extraction method may interfere with PCR amplification.

Commercial kits are methods that are increasingly used in present practice, due to their uncomplicated procedure and better quality of extracted DNA. The QIAamp DNA Stool Mini Kit is the most common commercial DNA extraction kit used with stool samples. The activity of cell lysis in the kits is performed by a patented buffer, an inhibitor absorber, and a silica gel membrane to remove inhibitors. However, this method sometimes gives a low DNA yield and fails to detect pathogenic DNA in stool samples [5,14]. It has been shown that a positive rate in the QIAamp DNA isolation kit is only at 48% for the detection of the 1.1 kbp SSU rRNA gene of *Blastocystis* sp. [5]. The QIAamp Fast DNA Stool Mini Kit contains InhibitEX buffer, which is specially formulated to separate the inhibitory substances and DNA-degrading substances from the DNA in stool samples, before binding the DNA to the silica membrane in the spin column. The QIAamp PowerFecal Pro DNA Kit, a new-generation kit that has recently been manufactured by Qiagen, comprises improved inhibitor removal technology and an added glass bead-beating procedure. Therefore, the cell lysis step in this kit is performed via both mechanical and chemical methods. It is claimed to be an effective method to remove PCR inhibitors and to improve DNA recovery from stool samples, even in the most difficult stool types. Moreover, it has been reported that DNA extraction from stool samples by the QB method can identify a greater bacterial diversity in microbiome studies, which is comparable with those of the QIAamp DNA Stool Mini Kit with a bead-beating step [11,12]. However, the inappropriate or too strong lysis method can cause the degradation of extracted DNA and fail to produce reliable PCR results [23]. The efficacy of this method for extracting DNA from several types of intestinal parasites is still questionable.

In this study, we assessed effective methods for extracting DNA from different kinds of intestinal parasites and removing of PCR inhibitors in stool samples. We compared the most common DNA extraction methods from stool samples. Pretreatment steps with glass bead-beating were also added to compare whether the yield of DNA extraction has been improved.

In the present study, the conventional P method was proved to provide a better DNA yield compared to the kit methods. However, only 7 samples of *S. stercoralis* were positive for PCR when using DNA extracted by the conventional P method. This might be due to the greater number of cells in the rhabditiform larvae of *S. stercoralis*. A much larger quantity of DNA was then extracted from the larvae, compared to the DNA extracted from the eggs and cysts of parasites. Adding a bead-beating step has been reported to improve DNA recovery, particularly when the fecal egg counts are high [24]. However, our study showed that the addition of a bead-beading step resulted in an increase in DNA yield from the P method, but not from the Q method (Figure 1A–C). This might be due to the maximum DNA binding capacity in the QIAGEN column. From the handbook, DNA yield from the Q method ranges from 5 to 100 μg, and the DNA concentration is typically 50–250 ng/μL. Similar to the previous study [24], we did not find a significant correlation between the DNA concentration of extracted DNA by each method; the even samples were well mixed to obtain a homogeneous mixture of the eggs in the samples. However, the DNA samples with a low concentration (<100 ng/μL) showed a non-significantly higher DNA concentration extracted by Q than by the P method (Figure 1D).

For the purity of the extracted DNA, the phenol-chloroform technique provided sub-optimal A260/280 ratios (geometric mean = 1.66), indicating that there are some protein contaminations in the extracted DNA. Moreover, the A260/230 ratios of DNA extracted by the phenol-chloroform technique were also very low (geometric mean = 0.58). The A260/230 ratio indicates the presence of organic contaminants, such as phenol, carbohydrate, glycogen, residual guanidine (often used in column-based kits), TRIzol, chaotropic salts, and other aromatic compounds. Therefore, extracting DNA by the phenol-chloroform technique showed a significant amount of the contaminants that would interfere with downstream applications.

The addition of the glass bead-beating step to the P method could increase the DNA yield, but not increase the purity of the DNA. The A260/280 and A260/230 ratios between P and PB were not significantly different (*p* > 0.05) (Figure 2A and Figure 3A). As expected, DNA purity, measured by both A260/280 ratios and A260/230, was found to be optimal in the Q and QB methods. These showed a sufficient quality of DNA for further study. The Q method produced pure DNA, in accordance with previous studies [6,11]. However, PCR amplification failed in 58.8% of the extracted DNA samples. Using DNA extracted by the Q method, none of *A. lumbricoides* and only 1 sample of *T. trichiura* was positive for PCR (Table 2). *A. lumbricoides* and *T. trichiura* eggs are known to be difficult to lyse. Additional egg disruption techniques are required to extract the DNA from these parasites [10]. Because *A. lumbricoides* eggs are resistant to many treatments, they are commonly used as an indicator organism for monitoring the efficiency of sludge and wastewater treatment [25]. Nevertheless, the sensitivity of commercial kits is still low when compared to the cultivation method since the previous reports showed that using the Q method alone cannot completely remove the inhibitor from the PCR amplification process [5].

To enhance the sensitivity of PCR amplification, suitable PCR enhancers, such as BSA, DMSO, and low molecular weight amides might be applied with suitable types of amplicons and primers [20,26]. The concentration of PCR enhancers must also be within the optimal levels to maximize the reaction [26,27]. In our previous study [15], we extracted the DNA of *Blastocystis* sp. from stool samples by using the E.Z.N.A.^®^ Stool DNA Kit (OMEGA Bio-tek Inc., Norcross, GA, USA), which has been shown to yield a comparable level of DNA by the Q method [28]. However, the sensitivity of PCR for the detection of *Blastocystis* sp. in the previous study [15] is higher than in the present study. This might be due to the addition of BSA into the PCR mixture to enhance the PCR reaction [15].

The addition of a bead-beating step has been reported to improve DNA recovery [11,24]. The cell lysis step in the QB method involves both mechanical and chemical methods. Unfortunately, we did not find a significant difference in DNA yield compared to using the Q method (Figure 1A,C). However, using the QB method resulted in a higher detection rate of PCR (Figure 6). Similar to the previous studies [11,12], this method can identify a greater diversity of organisms in the stool samples. All the parasites we tested were detected (Table 2). The QB method was also the most effective at removing PCR inhibitors, resulting in successful amplification in 94.1% of samples (Table 3). Only 5 DNA samples (5.9%) were still negative for PCR after spike tests.

Besides the bead-beating step, other methods have been reported to facilitate PCR amplification and remove the inhibitors. Similar comparative studies with other parasites and with other commercial kits have been reported to produce a high DNA yield and are effective for parasite detection in stool samples [4,5,6,8,9,10,11,12,13,14]. The Nucleospin DNA Stool kit (Macherey-Nagel) has been reported to be superior to the E.Z.N.A^®^ Stool DNA kit and QIAamp Fast DNA Stool kit for the detection of the ITS2 of *Haemonchus contortus* in fresh and frozen stool samples [28]. For the detection of 1.1 kbp of the SSU rRNA gene of *Blastocystis* sp., the ZR Fecal DNA kit from Zymo Research has provided the highest PCR positive rate (94%) compared to the QIAamp DNA Stool Mini Kit and MagNa Pure LC DNA Isolation Kit [4]. Similarly, the ZR Fecal DNA kit, combined with qPCR, shows the highest sensitivity for the identification of *Echninococcus multilocularis* DNA in spiked stool samples among 4 commercially available kits [5]. In this study, we compared the conventional DNA extraction methods with the most commonly used DNA extraction kits (Q and QB) and focused on the bead-beating pretreatment. Further studies on the other method principles and on other commercially available kits with a higher number of samples for each parasite will be useful for enabling wider practical applications. 

## 5. Conclusions

The QB method is a highly efficient DNA extraction method, not only for the molecular diagnosis of intestinal parasites but also for epidemiological study. Selecting the most suitable DNA extraction method facilitates the mapping and monitoring of disease, unless the disease burden has been underestimated. The present study also emphasizes the need for internal controls to detect PCR inhibitors when stool samples are used for parasitic detection using PCR.

## Figures and Tables

**Figure 1 diagnostics-12-02588-f001:**
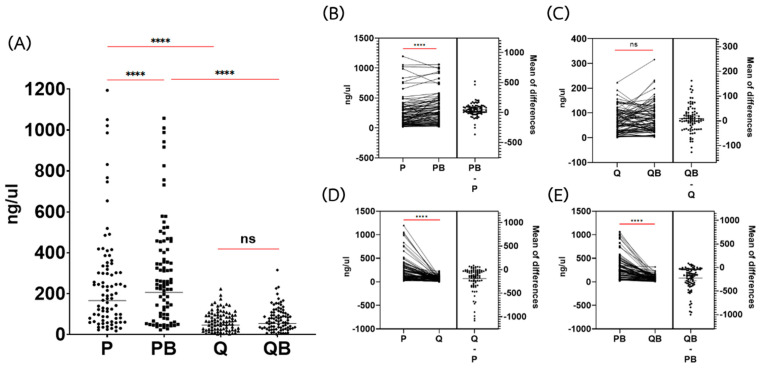
Comparison of the concentration of genomic DNA extracted via the phenol-chloroform extraction method (P), the phenol-chloroform extraction method with glass beads (PB), the QIAamp Fast DNA Stool Mini Kit (Q), and the QIAamp Power Fecal Pro DNA Kit (QB). (**A**) Dot plot of DNA concentration by various extraction methods. Each dot indicates the DNA concentration of each sample. (**B**–**E**) Estimation plot of DNA concentrations among the extraction methods. Each line in the left panel indicates the comparison of DNA concentration in each sample, extracted by each method. Each dot in the right panel indicates the difference between the DNA concentrations of each sample. The bar indicates the mean of the difference between concentrations. **** Statistical differences using Wilcoxon matched-pairs signed-rank tests (*p* < 0.0001), ns: non-significantly different (*p* > 0.05).

**Figure 2 diagnostics-12-02588-f002:**
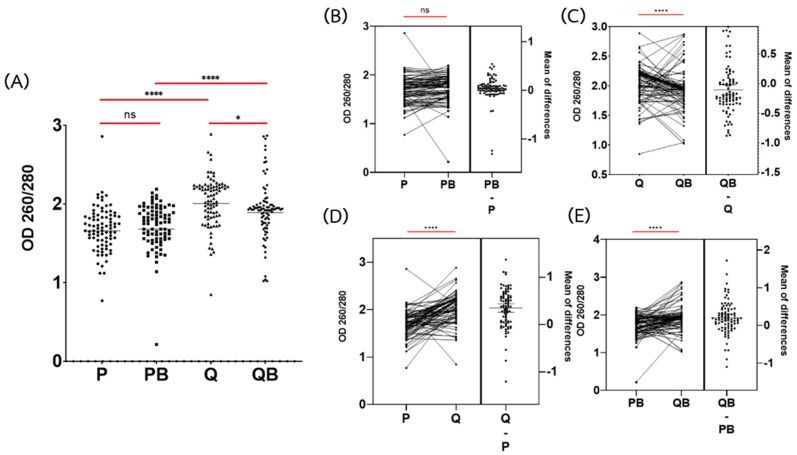
Comparison of the OD260/280 absorbance ratios of the genomic DNA extracted via the phenol-chloroform extraction method (P), the phenol-chloroform extraction method with glass beads (PB), the QIAamp Fast DNA Stool Mini Kit (Q), and the QIAamp Power Fecal Pro DNA Kit (QB). (**A**) Dot plot of DNA concentrations via extraction methods. Each dot indicated the DNA concentrations of each sample. (**B**–**E**) Estimation plot of the DNA concentrations among the extraction methods. Each line in the left panel indicated a comparison of the DNA concentration of each sample extracted by each method. Each dot in the right-hand panel indicated the differences between the DNA concentrations of each sample. The bar indicated the mean of the differences between concentrations. **** Statistical differences using a paired *t*-test (*p* < 0.0001), * Statistical differences using a paired *t*-test (*p* < 0.05), ns: non-significant differences (*p* > 0.05).

**Figure 3 diagnostics-12-02588-f003:**
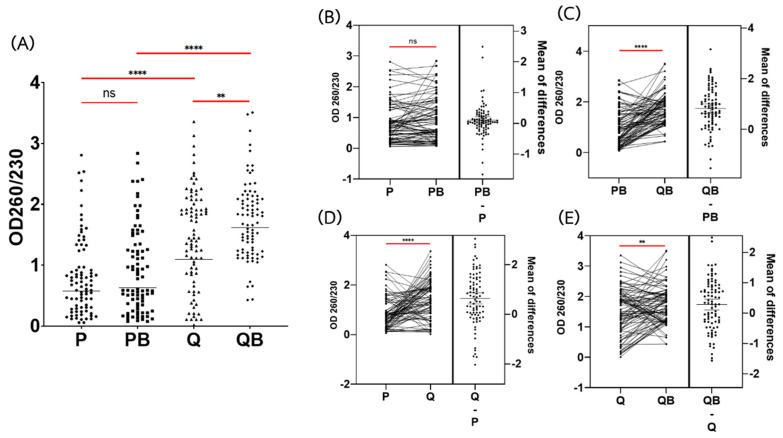
Comparison of the OD260/230 absorbance ratios of genomic DNA extracted by the phenol-chloroform extraction method (P), the phenol-chloroform extraction method with glass beads (PB), the QIAamp Fast DNA Stool Mini Kit (Q), and the QIAamp Power Fecal Pro DNA Kit (QB). (**A**) Dot plot of DNA concentrations using the extraction methods. Each dot indicates the DNA concentration of each sample. (**B**–**E**) Estimation plot of DNA concentration among the extraction methods. Each line in the left panel indicates the comparison of DNA concentration of each sample extracted by each method. Each dot in the right panel indicates the differences among the DNA concentrations in each sample. The bar indicates the mean of the differences between concentrations. **** Statistical differences using a paired *t*-test (*p* < 0.0001), ** Statistical differences using a paired *t*-test (*p* < 0.01), ns: non-significant differences (*p* > 0.05).

**Figure 4 diagnostics-12-02588-f004:**
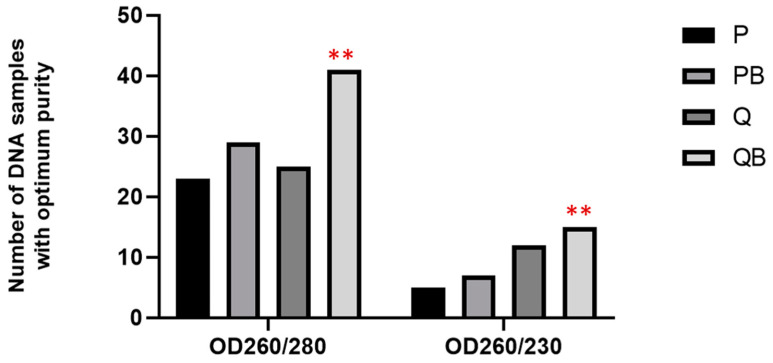
Comparison of the number of samples with an optimum purity of genomic DNA, extracted by the phenol-chloroform extraction method (P), the phenol-chloroform extraction method with glass beads (PB), the QIAamp Fast DNA Stool Mini Kit (Q), and the QIAamp Power Fecal Pro DNA Kit (QB). ** Statistical differences using a two-way ANOVA (*p* < 0.01).

**Figure 5 diagnostics-12-02588-f005:**
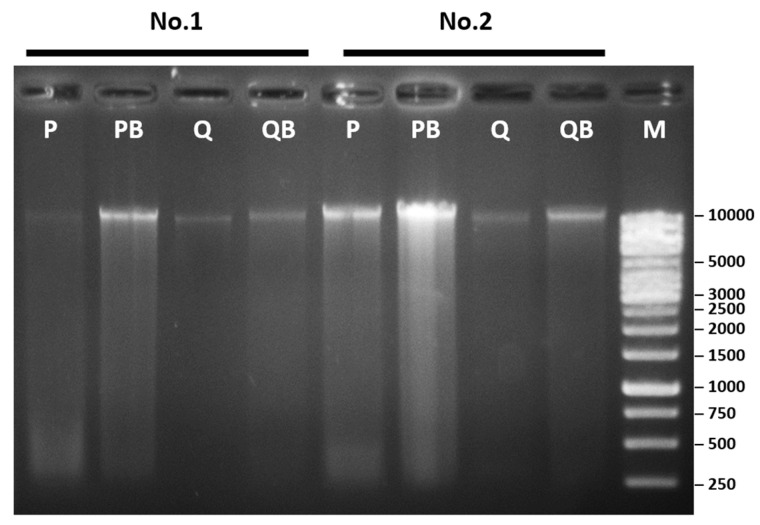
Evaluation of DNA integrity via the gel-electrophoresis of DNA extracted using the phenol-chloroform extraction method (P), the phenol-chloroform extraction method with glass beads (PB), the QIAamp Fast DNA Stool Mini Kit (Q), and the QIAamp Power Fecal Pro DNA Kit (QB).

**Figure 6 diagnostics-12-02588-f006:**
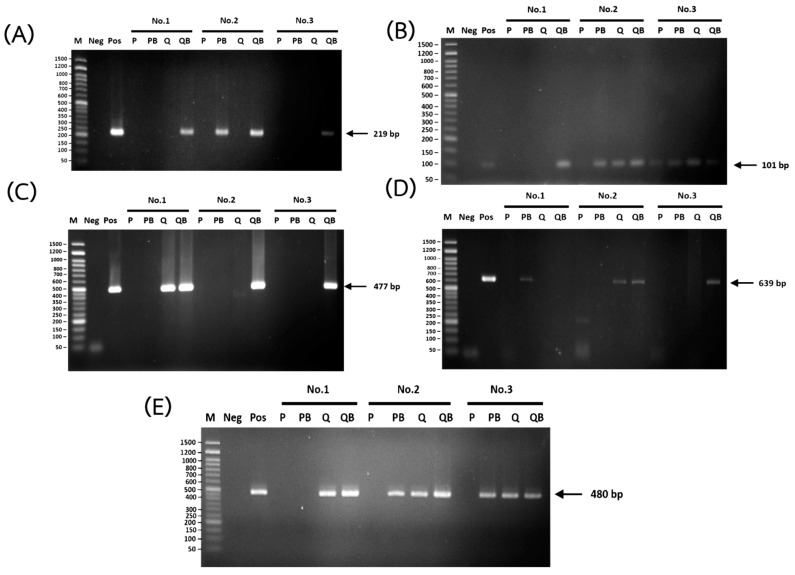
Representative results from PCR for the detection of *A. lumbricoides* (**A**), *S. stercoralis* (**B**), *N. americanus* (**C**), *T. trichiura* (**D**), and *Blastocystis* sp. (**E**) using DNA extracted from stool samples from each patient using the phenol-chloroform method (P), the phenol-chloroform method with glass beads (PB), the QIAamp Fast DNA Stool Mini Kit (Q), and the QIAamp Power Fecal Pro DNA Kit (QB). M: DNA marker; Neg: negative control; Pos: positive control.

**Figure 7 diagnostics-12-02588-f007:**
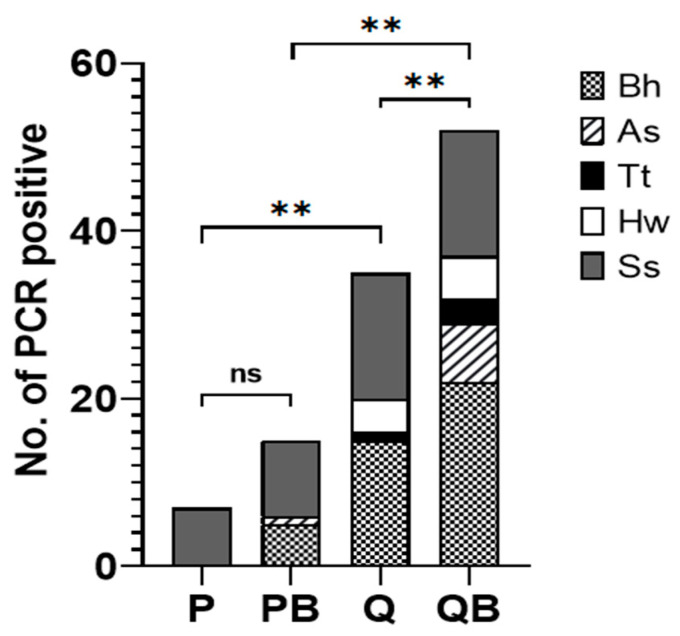
Comparison of the DNA extraction methods for the detection of intestinal parasites. Bh: *Blastocystis* sp., As: *Ascaris lumbricoides*, Tt: *Trichuris trichiura*, Hw: hookworm, Ss: *Strongyloides stercoralis*. ** Statistical differences using chi-square test (*p* < 0.01), ns: non-significant differences (*p* > 0.05).

**Table 1 diagnostics-12-02588-t001:** Sequences for primers for detections of *Ascaris lumbricoides*, *Necator americanus*, *Trichuris trichiura*, *Strongyloides stercoralis*, and *Blastocystis* sp. by PCR. F: Forward, R: Reverse.

Parasite	Target Region (Accession No.)	Primer (5′-3′)	Length (bp)	Product Size (bp)	Reference
*A. lumbricoides*	ITS1 (AJ000895.1)	F: GGT GAT GTA ATA GCA GTC GG R: TTC TCT CCA CCT TTC ATC G	20 19	219	[16]
*N. americanus*	18S rDNA (AF217891.1)	F: AGC ATT GCT TGA ATG CC R: AAG TAC CGT TCG ACA AAC AG	17 20	477	[16]
*T. trichiura*	18S rDNA (GQ352548.1)	F: CCG GGA AAC CAA AGT GTT TC R: GTA CAA AGG GCA GGG ACG TA	20 20	639	-
*S. stercoralis*	18S rDNA (AF279916.2)	F: GAA TTC CAA GTA AAC GTA AGT CAT R: TGC CTC TGG ATA TTG CTC AGT TC	24 23	101	[16]
*Blastocystis* sp.	18S rDNA (AB070989.1)	F: GGA GGT AGT GAC AAT AAA TC R: TGC TTT CGC ACT TGT TCA TC	20 20	480	[15,17]

**Table 2 diagnostics-12-02588-t002:** Comparison of the PCR positive rate for the detection of intestinal parasites, according to the DNA extraction method. Bh: *Blastocystis* sp., As: *Ascaris lumbricoides*, Tt: *Trichuris trichiura*, Hw: hookworm, Ss: *Strongyloides stercoralis*.

Method	PCR Positive (%)					
	Bh (*n* = 37)	As (*n* = 9)	Tt (*n* = 5)	Hw (*n* = 14)	Ss (*n* = 20)	Total (*n* = 85)
P	0	0	0	0	7 (35%)	7 (8.2%)
PB	5 (13.5%)	1 (11.1%)	0	0	9 (45%)	15 (17.6%)
Q	15 (40.5%)	0	1 (20%)	4 (28.6%)	15 (75%)	35 (41.2%)
QB	22 (59.5%)	7 (77.8%)	3 (60%)	5 (35.7%)	15 (75%)	52 (61.2%)

**Table 3 diagnostics-12-02588-t003:** Comparison of positive PCR rates for the detection of intestinal parasites before and after the spike tests. Bh: *Blastocystis* sp., As: *Ascaris lumbricoides*, Tt: *Trichuris trichiura*, Hw: hookworm, Ss: *Strongyloides stercoralis*.

Method	PCR Positive (%)											
	Bh (*n* = 37)		As (*n* = 9)		Tt (*n* = 5)		Hw (*n* = 14)		Ss (*n* = 20)		Total (*n* = 85)	
	PCR	PCR + Spike	PCR	PCR + Spike	PCR	PCR + Spike	PCR	PCR + Spike	PCR	PCR + Spike	PCR	PCR + Spike
P	0	2 (5.4%)	0	2 (22.2%)	0	0	0	11 (78.6%)	7 (35%)	10 (50%)	7 (8.2%)	25 (29.4%)
PB	5 (13.5%)	8 (21.6%)	1 (11.1%)	2 (22.2%)	0	1 (20%)	0	11 (78.6%)	9 (45%)	12 (60%)	15 (17.6%)	34 (40%)
Q	15 (40.5%)	22 (59.5%)	0	9 (100%)	1 (20%)	1 (20%)	4 (28.6%)	12 (85.7%)	15 (75%)	20 (100%)	35 (41.2%)	64 (75.3%)
QB	22 (59.5%)	34 (91.9%)	7 (77.8%)	9 (100%)	3 (60%)	5 (100%)	5 (35.7%)	13 (92.9%)	15 (75%)	19 (95%)	52 (61.2%)	80 (94.1%)

## Data Availability

Not applicable.

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
