# Peer review of "Comparative Study of DNA Extraction Methods for the PCR Detection of Intestinal Parasites in Human Stool Samples"

_diagnostics, 2022, doi:10.3390/diagnostics12112588_

Round 1

Reviewer 1 Report

The manuscript “Comparative study of DNA Extraction Methods for the PCR Detection of Intestinal Parasites in Human Stool Samples” evaluated the use of different DNA extraction methods for PCR detection of intestinal parasites. The subject is interesting and worth exploring, as clinical diagnosis of infection is often problematic, and conventional methods lack sensitivity and often take time to deliver results. Rapid detection of infection by PCR may be a good solution for routine diagnosis. The study is generally well constructed, however several methodological issues should be addressed, such as the choice of method for statistical analysis and spiking procedure. I suggest the manuscript may be published in this journal after moderate revision.

Major comments:

-          The introduction provides a description of currently used methods for the diagnosis of intestinal helminths. This description is a bit scattered and does not include egg count (fecal floatation) tests, serological tests and antigen tests.  The authors only discuss the methods used in the current study. Please give a brief description of the methods available, and explain the choice of the methods used in this study.

I recommend: Khurana, S. and Sethi, S., 2017. Laboratory diagnosis of soil transmitted helminthiasis. Tropical parasitology7(2), p.86.

-          The samples analyzed were of clinical patients and were diagnosed using different methods (either direct microscopy, ESCT or culture). This may suggest differences in parasite loads (as these methods differ in sensitivity). Was this issue addressed when analyzing the PCR results? Could it be that PCR fails to detect samples with low parasite burden (that could only  be detected in culture).

This issue may also affect PCR results since DNA extraction is performed on small sample size (200 mg). If parasitemia is low or egg shedding intermitted (as often the case), than the quantity of parasite DNA may be different in different aliquots of the stool sample.

-          DNA spiking evaluates the threshold of the quantity of parasite DNA detected by the PCR assay. 1) adding known amounts of plasmid DNA of each parasite to fecal DNA extracted from negative samples (creating a calibration curve) will provide better evaluation of assay sensitivity (which will include the presence of any DNA inhibitors. 2) If spiking was performed on samples already containing an unknown amount of parasite DNA, than the conclusions regarding the presence of inhibitors is limited (since sufficient amount of specific DNA can sometimes overcome the presence of such inhibitors).

-          Please make sure that P-values are stated for all described comparisons.

-          Why were the DNA quantity and quality (260/280 and 260/230) means only compared in pairs using non-parametric test, and not all groups compared by ANOVA and post-hoc tests?

-          Both 260/280 and 260/230 ratios have optimal values for pure DNA. However, the authors analyzed the data by comparing “high” or “low” values between methods. Moreover, some deviation from the optimum is acceptable and some suggest impurity. The data should be analyzed in comparison to these optimal values, and not as absolute numbers.

-          The fact that PCR detection was less sensitive than microscopy is surprising, and may be due to intermitted excretion of parasites or eggs or to low parasitemia. More sensitive molecular methods such as nested PCR or real-time PCR may increase the sensitivity of detection.

-          Large parts of the discussion either describe previous knowledge (which belong in the introduction) or repeat the results. I suggest trying to address the results in light of the previous knowledge (compare the findings to previous studies, offer possible explanations to unexpected results). Also, please address study limitations and give practical conclusions for future diagnostics.

Specific comments:

-          Line 38 – please replace “detections” with “detection methods”.

-          Lines 39-40 – please either add “and” between microscopy and ESCT or add methods to the list. This list is very short. These are only two techniques of direct microscopy, what about fecal floatation methods and egg counts?

-          Please provide the manufacturer’s name and address for both extraction kits used.

-          Lines 164-170 – please provide a more detailed description of the spiking process (the plasmid copy number(s) used, the positive controls, etc.).

-          Table 1 – 1) please provide citations for the primers designed prior to this study. 2) 100 bp is quite short for conventional PCR. Has this method been validated previously?

-           Line 187 – please delete “by” before “the Q method”.

-          Lines 187-188 – please provide a P-value.

-          Line 192 – 1) please delete “the” before “low DNA conc.” 2) was the DNA concentration low in both methods or just in P?

-          Figure 1 (and also 2 and 3) – please provide a description of the data presented in (A), (B), (C), (D) and (E) in the figure legend.

-          Line 246 – please delete “the” before “comparing”.

-          Table 2 and figure 5 – please provide an explanation of the abbreviations in the figure legends.

-          Figure 5 (and also 1, 2 and 3) – at appears that not all methods were compared to each other (only P-Q, P-PB and Q-QB) – was QB superior to all other methods? Or just Q?

-          Lines 288, 291 – please add P-values to these comparisons.

-          Line 310 – please delete “in the” before infants. (why was this stated? were all participants in this study adults? This was not stated).  

-          Lines 302-313 – DNA extraction is not only affected by inhibitors, but also mechanical properties such as fibers etc. in the stool and think membranes or walls of parasites and oocysts. Also, when mentioning inhibitors, both DNA inhibitors and PCR inhibitors should be considered (as these are not the same).

-          Line 314 – please delete “been”.

-          Line 318 – did any of the used methods include such steps?

-          Line 322 – please rephrase to: “But their efficiency varies and is far from the complete removal of inhibitors”.

-          Lines 302-350 – belong more in the introduction.

-          Lines 352-356 – was this study aimed to prove that QB is superior to other methods? Or to compare several available methods? Was this study supported by the manufacturer in any way? Is so, this should be stated.

-          Throughout the text the abbreviations P, PB, Q and QB were used. However, in the discussion the abbreviations and full names are both used. Please unify.

Reviewer 2 Report

Line 6 -> No email to contact Siriporn Sirirungruang Line 7 -> "lumbricoides, Trichuris trichiura, Hookworm, and Strongyloides stercoralis" I suggest adding the following statement to the phrase: which was confirmed by such and such means Line 130 -> what is an "ATE buffer"? If this is a typo, you need to correct Line 181 -> "T. trichiura" - give the full Latin name of the species - Trichuris trichiura Line 203, 222 and 243 -> please decipher the ns abbreviation (Figure 1a) in the header of Figure 1 Line 222 -> is not deciphered what "*" means (Figure 2a) Line 207, 226 and 243->  ” test. (P<0.0001) ” – the dot after the word test is probably unnecessary Line 239 -> is not deciphered what "**" means (Figure 3a and 3e) Line 276-277 -> it is necessary to give in the title of the Table the decoding for Bh, As, Tt, Hw, Ss Line 278 -> it is necessary to give in the title of the Figure 5 the decoding for Bh, As, Tt, Hw, Ss Line 300 -> it is necessary to give in the title of the Table 3 the decoding for Bh, As, Tt, Hw, Ss

Reviewer 3 Report

This is an interesting article on the comparison of various methods to extract DNA of parasites, both protozoan and nematodes, and to analyze the quality of the DNA extracted.

In general, the manuscript is of interest to the potential readers of the journal and to parasitologists in general, but particularly to those using molecular techniques in parasite diagnostics.

However, there are various aspects which could improve the quality of this research.

The description of the phenol-chloroform technique is correct, and can be followed by other scientists, but it is not the case of the both QUAamp techniques. The authors should describe these techniques in more detail, and not only refer to the manufacturer’s instructions.

On the other hand, and the most important questions are the number of samples analysed, not in total but concerning each parasite species. Of the total 85 stool samples analyzed, only 9 samples containing Ascarislumbricoides and 5 containing Trichuris trichiura were analyzed. This is, comparatively, a very small sample size, especially when taking into account that the eggs the two nematodes are resistant and the DNA extraction is too difficult. This sample size of both species should be larger, with at least 30 samples for each species. The same goes for the hookworms and Strongyloides, although, in these cases, the sample is greater.

Moreover, the only parasite of which a sufficient number of samples was analyzed is Blastocystis sp. (not spp.). Nevertheless, in my opinion there are various shortcomings concerning the choice of this parasite species: 1) the DNA extraction of Blastocystis is too easy compared to other protist parasites, such as CryptosporidiumMicrosporidia, etc., and 2) the DNA identification of Blastocystis frequently results in false negative results. Thus, the authors should have chosen another protist species such as those mentioned above.

Although the statistical analysis has been carried out correctlythe authors use the Wilcoxon testThis test canbe used when the sample does not follow a normal distribution or does not have any of the followingassumptionshomoscedasticitylinealitynon-multicollinearity and no autocorrelation; i.ethis is a non-parametric test used to compare paired samplesButactually, did the authors testbefore the use of theWilcoxon test, if the samples have a non-normaldistribution or did not follow any of the above-mentionedassumptions?

Reviewer 4 Report

The reviewed manuscript is dedicated to comparison of various DNA purification methods from stool samples. Diagnostics of intestinal parasites often relies on microscopy or culture. However, these approaches could be time-consuming and be low sensitive. PCR-based testing is a fast and reliable way to detect various parasites. However, PCR needs purified DNA template, while DNA obtained from stool samples is often contaminated by various inhibitors. Also, DNA amount could be insufficient due to durable shells of parasites. Thus, a careful choice of DNA isolation methods is necessary for reliable testing. In that sense, the results presented in the manuscript are timely and interesting for the molecular diagnostics of parasites. However, a few comments need to be made concerning several topics.

Major issues:

1.       Authors are encouraged to provide more detailed review of similar studies dedicated to the similar topic, e.g., comparison of DNA isolation methods from stool samples.

2.       Spiking of stool samples with plasmid DNA would help to determine DNA recovery. This parameter would help to evaluate the efficacy of DNA purification methods.

3.       The length of amplicons can be crucial for reliable PCR testing. Several amplicons have a length at the range of 477-679 bp. It could reduce the efficiency of PCR, particularly, in the presence of inhibitors.

4.       Experiments with several technical repeats of the same sample purified by the same methods would be useful to clarify reproducibility of DNA isolation methods.

5.       Authors are encouraged to present gel-electrophoresis of DNA samples to evaluate integrity and length of purified DNA fragments.

6.       Was there a correlation between inhibition rate and the negative status of P and PB samples?

7.       DNA titration in PCR would reduce inhibitor load and would increase the PCR efficacy.

Minor issues:

1.       Authors are encouraged to present more detailed legend for the Figures 1, 2, particularly, designation of content on separate panels.

2.       Amount of plasmids in spiking experiments is not specified.

Round 2

Reviewer 3 Report

The manuscript has been improved and the authors have generally addressed the points raised in my previous review.

Consequently, the Ms can be accepted in its present form.

Reviewer 4 Report

The authors answered clearly on all raised questions and no further corrections nned to be introduced to the manuscript.